# Novel Techniques Targeting Fibroblasts after Ischemic Heart Injury

**DOI:** 10.3390/cells11030402

**Published:** 2022-01-25

**Authors:** Sirin N. Cakir, Kaitlin M. Whitehead, Hanifah K. L. Hendricks, Lisandra E. de Castro Brás

**Affiliations:** Department of Physiology, The Brody School of Medicine, East Carolina University, Greenville, NC 27834, USA; cakirs18@students.ecu.edu (S.N.C.); whiteheadka20@ecu.edu (K.M.W.); hendricksh19@students.ecu.edu (H.K.L.H.)

**Keywords:** cardiac fibroblast, myocardium, novel delivery methods, nanoparticles, fibroblast-specific

## Abstract

The great plasticity of cardiac fibroblasts allows them to respond quickly to myocardial injury and to contribute to the subsequent cardiac remodeling. Being the most abundant cell type (in numbers) in the heart, and a key participant in the several phases of tissue healing, the cardiac fibroblast is an excellent target for treating cardiac diseases. The development of cardiac fibroblast-specific approaches have, however, been difficult due to the lack of cellular specific markers. The development of genetic lineage tracing tools and Cre-recombinant transgenics has led to a huge acceleration in cardiac fibroblast research. Additionally, the use of novel targeted delivery approaches like nanoparticles and modified adenoviruses, has allowed researchers to define the developmental origin of cardiac fibroblasts, elucidate their differentiation pathways, and functional mechanisms in cardiac injury and disease. In this review, we will first characterize the roles of fibroblasts in the different stages of cardiac repair and then examine novel techniques targeting fibroblasts post-ischemic heart injury.

## 1. Introduction

Cardiac fibroblasts are one of the most abundant cell populations, in terms of cell numbers, in the heart [1]. Cardiac fibroblasts are found in the interstitial and perivascular matrix and contribute to the structural, mechanical, biochemical, and electrical properties of the myocardium [2]. Traditionally, fibroblasts are defined as cells of mesenchymal origin, and views on their function involve the production of extracellular matrix (ECM), source of fibrosis, ECM remodeling, tissue repair, and scar formation. Aside from their role in matrix remodeling and scar formation, one of their most striking characteristics is their cellular plasticity. Based on the local tissue microenvironment, the phenotypically plastic fibroblast goes through a series of changes to acquire a required function. This plasticity means fibroblasts are not terminally differentiated cells with a defined phenotype and function. Therefore, these cells do not seem to express specific markers, which for a long time, has been a limiting factor in studying fibroblasts in vivo. As fibroblasts are actively involved in several pathologic processes, including inflammatory response, vasculogenesis, and tumorigenesis [3], they contribute to the onset and progression of many cardiovascular diseases.

The adult human heart has low regenerative capacity; thus, myocardial repair depends on the removal of dead/damaged cardiomyocytes and the formation of a scar. After an injury (e.g., ischemia), heart tissue healing includes a chain of events that start with clearance of the injured site from dead cells and cell debris followed by ECM deposition. To prevent abnormal remodeling, myocardial repair needs to be precisely controlled to avoid cellular overactivity, which would lead to aberrant healing and organ dysfunction. After an ischemic event, such as myocardial infarction (MI), cardiac fibroblasts undergo phenotypic changes allowing them to participate in post-MI inflammatory response and repair, by secreting both inflammatory molecules, necessary for the resolution of inflammation and the ECM components which will form the myocardial scar [4]. The rapid differentiation of cardiofibroblasts into myofibroblasts (myoFB) and their subsequent proliferation are required to maintain the structural integrity of the ventricular walls following injury [5].

In this review, we will first characterize the roles of fibroblasts in the different stages of cardiac repair and then examine novel techniques targeting fibroblasts post-ischemic heart injury.

## 2. Cardiac Fibroblast Roles and Subpopulations

Fibroblasts have been primarily thought of in terms of how they utilize their extensive endoplasmic reticulum to secrete and maintain the ECM scaffold that supports adjacent cardiomyocytes [6]. However, the ECM network plays a crucial role in cardiac homeostasis, not only by providing structural support, but also by facilitating force transmission and by transducing key signals to cardiomyocytes, vascular cells, and interstitial cells [7]. Since fibroblasts are the main producers of cardiac ECM, this means fibroblasts are essential for all ECM roles. For instance, fibroblasts are critical for preserving normal pump function and regular electrical activation of the heart, since cardiomyocytes can only maintain normal contractility if embedded in a suitably configured ECM that preserves a regular three-dimensional structure. In the intact myocardium, fibroblasts form a dense cellular network around cardiomyocytes with individual fibroblasts being precisely squeezed in between densely packed cardiomyocytes [8].

Ischemic injury both may lead to cardiomyocyte death and cause dynamic changes in the cardiac ECM that contribute to the regulation of the inflammatory and reparative phases [7]. Compared to cardiomyocytes, cardiac fibroblasts are considerably less sensitive to acute hypoxic conditions induced by ischemia. To that fact, fibroblasts are one of the first cardiac cells to sense and respond to heart injury, this is likely due to their abundant presence in the heart interstitium and close proximity to muscle cells [4]. A large body of experimental evidence supports a role for cardiac fibroblasts in inflammatory activation and responses [9,10]. However, the actual impact of fibroblasts to the acute post-injury inflammation is yet elusive, burdened by the contribution of other resident non-immune cells, such as vascular endothelium and mast cells, which can produce a broad spectrum of proinflammatory factors and are involved in the post-MI inflammatory processes [11,12].

### 2.1. Cardiac Fibroblast Proinflammatory Actions after Ischemic Injury

MI is associated with a sterile inflammatory response [13]; this triggers immediate vasodilation and fluid leakage followed by leukocyte extravasation that can occur within minutes to hours [14]. The net result is rapid delivery of soluble defenses, such as antibodies, complement, and collectins, and cellular ones, such as granulocytes and monocytes [15]. Both granulocytes and monocytes (once differentiated) secrete inflammatory factors, including enzymes, that help to clear cellular debris and remove necrotic cells and tissue. Activation of matrix-degrading proteases is crucial to facilitate the migration of leukocytes in the site of injury and promote clearance of dead cells and ECM debris [16]. Endogenous matrix-degrading enzymes, such as matrix metalloproteinase (MMP)-9, promote fibroblast survival, inflammation, and tissue degradation [17]. Like many immune cells, fibroblasts express MMPs as well as a variety of pattern recognition receptors, including Toll-like receptors (TLRs) [18,19]. TLR-ligand activation can directly activate fibroblasts and promote their differentiation into myoFBs, which are more mobile and contractile with a greater synthetic ability to produce ECM proteins [20]. This has been demonstrated experimentally. Cultured cardiac fibroblasts stimulated with the pro-inflammatory factors interleukin (IL) 1β and tumor necrosis factor alpha (TNF-α), both of which are secreted upon TLR activation, displayed differentially regulated production of tissue inhibitors of metalloproteinases (TIMPs) and activated a disintegrin and metalloproteinase domain-containing protein 10 (ADAM10)—all-important inflammatory mediators that regulate myocardial remodeling [21]. Another way by which the fibroblast regulates the ischemic inflammatory response is by homing circulating leukocytes and promoting activation of the endothelium to allow leukocyte extravasation [22,23,24]. Fibroblasts accomplish this through the expression of cytokine-induced C-C and C-X-C motif chemokines such as monocyte chemotactic protein (MCP)-1, macrophage inflammatory protein (MIP)-1, CCL5 (chemokine C-C motif ligand 5), and CXCL10 [25,26]. Cardiac fibroblasts not only modulate the recruitment of immune cells, but also regulate their behavior, retention, and survival in the ischemic heart. Crosstalk between fibroblasts and leukocytes normally results from the interaction between the surface antigen cluster of differentiation (CD) 40 on fibroblasts and its ligand, CD40L, which is expressed on immune cells. This receptor-ligand binding interaction induces the up-regulation in fibroblasts of both the intercellular adhesion molecule (ICAM)-1 and the vascular cell adhesion molecule (VCAM)-1 [27,28], which in turn are important for the induction of chemokine production.

### 2.2. Cardiac Fibroblasts during Resolution of Inflammation and Repair

The acute inflammatory response should be dampened in a timely manner to avoid further exacerbation of cardiac inflammation and propagation of heart damage [29]. This requires the coordinated actions of several different cell types and involves the participation of the ECM. Following the suppression of the inflammatory response, fibroblasts and endothelial cells infiltrate the infarct, signaling the start of the proliferative phase. During this stage, fibroblasts acquire the proliferative myoFBs phenotype due to the induction of growth factor signaling that initiates the synthesis of contractile and ECM proteins, as well as the promotion of cell mobility. Specifically, the release of transforming growth factor beta (TGFβ), basic fibroblast growth factor, vascular endothelial growth factor (VEGF), platelet-derived growth factor, MMPs, and proteolytic enzymes change the microenvironment towards a pro-angiogenic and pro-migratory phenotype. The inflammatory response is reduced when the infarct is purged of dead cells and matrix debris. Neutrophils perish and are phagocytosed by macrophages, whereas lymphocytes and macrophages generate inhibitory mediators that dampen pro-inflammatory signals, such as interleukin 10 (IL-10) and TGFβ along with induction of intracellular inhibitors of innate immunity such as interleukin receptor-associated kinase M (IRAK-M). Both TGFβ and IL-10 are predominant inhibitors of chemokinetic signaling and therefore are important mediators of the resolution of inflammation; of note, several cells secrete these factors including fibroblasts [29,30]. IRAK-M serves as a functional decoy preventing excessive TLR/IL-1-mediated responses. IRAK-M suppresses IL-1 dependent signaling that leads to the prevention of adverse post-MI remodeling, down-regulation of inflammatory effects of leukocytes, and inhibition of the matrix-degrading activity of fibroblasts [31].

Interestingly, some inflammatory cytokines that cooperate in proinflammatory activation of cardiac fibroblasts can demonstrate opposed effects on the fibrotic fibroblast activity. For example, the negative regulation of TLR-signaling is necessary for the resolution of the post-ischemic inflammation [32]. As mentioned above, IRAK-M which is expressed by fibroblasts inhibits TLR signaling and may contribute to myoFB conversion [33]. The pro-inflammatory IL-1β and TNF-α may indirectly induce the profibrotic activity of myoFBs through increasing the expression of type 1 angiotensin II (AT1) receptors [34]. Moreover, CD44 expression is markedly upregulated in myoFBs [35,36]; CD44 is essential for the clearance of hyaluronan fragments, which are formed during the acute inflammatory response, and their clearance has been shown to be important for the resolution of inflammation [37].

### 2.3. Fibroblast Subpopulations

As mentioned above, the lack of fibroblast-specific markers has made the identification and characterization of fibroblast subpopulations challenging. Moreover, phenotypic alterations of subpopulations of fibroblasts have previously made the distinguishment between these populations somewhat complicated. For a while, these populations have been recognized from other myocardial interstitial cells based on their expression of discoidin-domain receptor 2, platelet-derived growth factor receptor alpha (PDGFRα), and transcription factor 21 (Tcf21) [38,39,40,41]. In recent studies, single-cell sequencing has greatly advanced fibroblast phenotyping by providing information that allows us to differentiate between similar cell types (Figure 1). The location of these cells as well as the microenvironment of residence for each cell type are key components for understanding these cell populations. The proximity of these cell types to blood vessels, other cell types, as well as the content of their ECM all play a role in how these cells function and the roles they carry out. A deeper understanding of these populations, their functions, and their origins are essential for refining studies involving cardiac ischemia diagnosis, treatment, and therapies.

Resident fibroblasts, also known as resting or quiescent fibroblasts, maintain the structural integrity of the matrix network and additionally regulate collagen turnover. This subpopulation exhibits strong proliferativity after MI and studies have shown that fibroblasts present in the infarcted scar originate from several other populations, besides resident cells. Recent studies have identified Tcf21 as a resident fibroblast marker [42].

Post MI, ECM proteins and matricellular macromolecules that enrich the matrix network provide activating signals to fibroblasts which then locally activate fibrogenic growth factors at the site of injury. This new cellular subpopulation—activated fibroblasts—is one of two known central cellular effectors in cardiac fibrosis [43,44]. The activated fibroblasts are not only a source of matrix proteins, they also aid in the regulation of matrix remodeling by producing proteases such as MMPs and their inhibitors TIMPs [45]. This subpopulation is demonstrated to have originated from resident fibroblast populations through experiments involving lineage tracing strategies, bone marrow transplantation experiments, and parabiosis models [39,40,46]. An important distinguishing factor for activated fibroblasts is their secretion of periostin as this is not present in resident fibroblast populations [45]. Many consider the activated fibroblast a transition state between a fibroblast and a myoFB once these cells have increased secretory profile, often as a response to mechanical stress, but do not show overexpression of contractile proteins—a hallmark of myoFBs [47,48].

Like the activated fibroblasts, myoFBs also produce matrix proteins making them another type of cardiac fibrosis cellular effector. Due to increased synthetic and secreting capacity, myoFBs develop a large endoplasmic reticulum network [49]. Determined majorly by their prominent endoplasmic reticulum as well as their expression of contractile proteins such as alpha smooth muscle actin (αSMA) this population tends to exhibit a range of phenotypic profiles [49,50]. For example, early myoFBs may not express αSMA but show a stress fiber network and form focal adhesions termed proto-myofibroblasts [48]. Nonetheless, it is generally accepted that αSMA is a marker for myoFBs [42]. Although myoFB is essential for myocardial healing after ischemia, the persistence of myoFB contributes to maladaptive remodeling and progressive decline of cardiac function.

Resulting due to myoFB differentiation is the subpopulation of matrifibrocytes. Described as a specialized matrix-producing cell, these cells were observed post-MI weeks after the fibroblasts’ proliferative stage. This subpopulation was identified after a study stimulating the fibroblasts after cardiac injury, where the fibroblasts making up the scar tissue were unresponsive to stimulus. Meanwhile, fibroblasts surrounding the injured area behaved normally [51]. Immunohistochemistry analysis demonstrated these cells express proteins Chad and Comp, both of which are associated with chondrocytes and osteoblasts [51]. Chondrocytes and osteoblasts are cells commonly found in bone, cartilage, and tendon-like tissues suggesting matrifibrocytes to be essential for the maintenance and stabilization of the mature scar. Although these are mainly present after the scar has matured, Comp was recently identified as a marker for this population [42].

## 3. Fibroblast-Specific Research Tools

### 3.1. Transgenic Mice

Mouse models have become increasingly popular due to their easily manipulated and well-characterized genome. Specifically, in studies involving diagnostics, prevention, and therapy procedures, mice have become an increasingly attractive model due to their short gestational periods, smaller size, larger litter size, low maintenance cost, and more animals being able to be used. Arguably, the most important reason for the use of mouse models in cardiac research is many studies have uncovered that the development of the heart and vasculature is being regulated by similar genes and signaling pathways in both mice and humans [42]. Aside from their ability to be used in surgeries and pharmacological testing, transgenic mice have become a valuable tool in post-MI studies. Transgenic mice contain either a loss of function or gain of function mutation resulting from artificially introduced genetic material to their genome [52]. One of the most popular tools in science today, used in combination with transgenic mice, is the Cre recombinase, a key instrument for lineage tracing and cell-specific mutations. Notably, for fibrosis studies, mice have been created to target specific fibroblast populations making mutations and reporters increasingly cell-specific. The identification of fibroblast markers has resulted in new ways to identify the origin of a cell and its descendants, also known as lineage tracing tools [53]. The principle of lineage tracing is that the expression of reporter genes under the control of the *cis*-regulatory elements of a cell-specific gene permits the identification of this cell population and its descendants. Thus, one issue that arises with creating lineage tracing tools is finding markers specific to the cell type of interest. The identification of fibroblast markers (described above) has facilitated the development of fibroblast lineage-tracing tools. Currently, there are two categories of lineage tracing tools, direct and indirect. Direct lineage tracing involves a tissue-specific promoter or locus controlling a reporter gene such as green fluorescent protein (GFP) [54,55]. These types of lineage tracing tools can be introduced to the genome either as knock-ins or transgenes. Indirect lineage tracing is where the Cre-loxP recombination system is used, where Cre is driven by a tissue-specific promoter or locus and the reporter is downstream of a stop codon and flanked by two loxP sites [56,57].

A popularly manipulated gene used to generate lineage tracing of fibroblasts in mice is Tcf21. Tcf21 is essential in cardiac fibroblast growth and maintenance making it an excellent tool for fibroblast-related studies. Oftentimes the Tcf21 mutation is used as a driver for GFP reporters. In previous studies, it was used as a driver for the collagen 1a1 enhancer-controlled GFP reporter as well as GFP that was inserted into the gene encoding PDGFRα [38,46,54]. Lines such as these have helped to determine where the fibroblasts originate from in development as well as the source of fibroblasts post-MI. In addition, Tcf21 presents itself in other studies as an indirect lineage tracing tool when combined with the Cre-loxP system. The line containing the Tcf21 and tamoxifen-inducible Cre was crossed with a line containing a GFP insert into the Rosa26 gene, allowing for the tamoxifen to activate the Cre in the Tcf21positive (+) cells marking them permanently as GFP positive. This tool coupled with visualization of BrdU or αSMA allowed to identify respectively proliferating fibroblasts within the post-MI scar area and determine whether these cells were adopting a myoFB phenotype [50]. Other fibroblast specific lineage tracing strains include Acta2-CreERT2 [57], Col1a2-CreERT [58], Tcf21-Cre [59], Tie2-Cre [60], among others.

Another popular mouse strain in fibroblast research is the tamoxifen-inducible Cre periostin gene-targeted mice. This line has been used for genetic lineage tracing in myoFBs, experimental deletion of periostin+ myoFBs and deletion of periostin+ activated fibroblasts after MI [59]. This mouse model has led to the discovery that reducing periostin+ myoFBs results in less collagen production and scar formation after MI [61]. In addition, lineage tracing has uncovered that these myoFBs revert to a less-activated state after injury resolution, and they’re derived from tissue-resident fibroblasts of the Tcf21 lineage [38].

A benefit of using transgenic mice is the ability to manipulate them further to create more advanced tools for additional experimentation. An example of this is provided in a study using the activating transcription factor 3 knock-out (ATF3KO) mouse [62]. Activating transcription factor 3 is highly expressed in cardiac fibroblasts as a response to hypertensive stimuli. In this study, the lab established a conditional cardiac fibroblast-specific ATF3 transgenic mouse as well as a miRNA-aided/lentivirus-mediated noncardiac myocyte ATF3 overexpression model. These lines allowed them to target and manipulate cardiac fibroblasts using a cell-specific line, as well as perform rescue experiments using the overexpression model. In doing so, they uncovered evidence to support ATF3 as a possible therapeutic agent against hypertensive cardiac remodeling [62]. It is not this review’s intention to describe a detailed list of all fibroblast-specific transgenic mouse models, only to provide an overview of the importance and benefit of using such a tool in cardiac research. For a detailed review of available fibroblast-specific mouse lines, we recommend reading Fu et al. [50].

### 3.2. Nanoparticles

Nanoparticle (NP) based drug delivery systems are new but swiftly developing; these systems can be employed to serve as means of diagnostic tools or to deliver therapeutic agents to definite sites in a controlled manner. NP-based drug delivery systems became an attractive carrier device in the cardiac field to target molecules playing roles in inflammation, vascularization/angiogenesis, and circulation [63,64,65,66,67,68]. NPs are appealing for their ability to remain in circulation for an extended period of time, for maintaining protein stability, and also for allowing controlled release of its payload [69].

NP-based drug delivery systems consist of inorganic and organic NPs. Inorganic NPs include gold nanoparticles (AuNPs) [70], mesoporous silica nanoparticles (to date, not yet used in cardiac research) [71], silver nanoparticles (AgNPs) [72], carbon nanotubes (CNTs) [73], and quantum dots (QDs) [74]. Organic NPs are considered biodegradable and biocompatible, examples include liposomes [75], polylactic-co-glycolic acid (PLGA) NP [76], and polymeric micelles [77]. Overall, the NPs flexibility and characteristics make them very attractive tools for drug delivery in many settings, including cardiac repair.

AuNPs have been gradually recognized as one of the most promising nanomaterials, this is attributed to their unique optical, electronic, sensing, and biochemical characteristics, including low phototoxicity [78]. Depending on the application, AuNPs diameters range from 1 to 100 nm. Several AuNPs have been developed to target fibroblasts in a variety of diseases, such as cancer, arterial injury, and cardiac injury [79,80,81]. Cy5.5-MMP-AuNPs were developed for near-infrared fluorescence imaging of MMPs expressed in MI, tumors, atherosclerosis, and other diseases [82]. AuNPs also have been used by cardiac tissue engineering research to create an AuNP-decellularized matrix hybrid; in this study, the authors demonstrated that AuNPs were able to significantly increase the conductivity of the natural material without affecting the mechanical properties relevant for cardiac tissue engineering [70].

AgNPs are particles composed of silver atoms and like AuNPs they usually range in size from 1 to 100 nm [83]. AgNPs are extensively applied for their broad-spectrum and splendid antibacterial abilities. Surface immobilization of AgNPs is one of the most effective ways to increase the antibacterial property of materials [84]. AgNPs have demonstrated marked efficacy when used in antimicrobial applications and wound healing activity by fibroblasts [85]. On the other hand, researchers have shown that pulmonary exposure to Ag core AgNP induces a measurable increase in circulating cytokines, expansion of cardiac ischemia-reperfusion (I/R) injury, and is associated with depressed coronary constrictor and relaxation responses [86]. The potential effects of NPs on targeted cell and tissue function, apart from cytotoxicity, are not completely understood. In an attempt to identify other roles of AgNP, fibroblasts were exposed to different concentrations of AuNPs and AgNPs; these chemotaxis assays showed NPs reduced fibroblast migration through transwell chambers, demonstrating that metal NPs may influence fibroblast function by negatively modulating the deposition of ECM and/or altering the expression of ECM receptors, cytoskeletal reorganization, and cell migration [72]. Similarly, AgNPs have been reported to alter epithelial basement membrane integrity, cell adhesion molecule expression, and TGFβ secretion [87], further supporting the potential of AgNP use as an anti-fibrotic therapy.

CNTs are an allotropic form of carbon with dimensions in the nanoscale and length in the micrometer scale that form a needle-like structure with a large surface area [88]. Researchers have demonstrated that CNTs complexed with chitosan, which is a hydrophilic polymer, result in a hydrogel. This material was used to investigate fibroblast deposition of collagen and ECM organization in vitro, followed by the study of the effects of these compounds on dermal wound healing using a mouse model with induced full-thickness wounds [89]. The authors demonstrated that fibroblasts were viable in the presence of the chitosan-CNT complexes, were able to effectively organize and contract the ECM, and improved the re-epithelialization of the healing wounds; however, they also were associated with increased fibrosis in vivo. This study cautions the use of CNTs as a fibroblast-targeting system. However, CNTs have been used successfully to improve the conductivity of biomaterials and reduce fibroblast proliferation when coculturing cardiomyocytes and cardiac fibroblasts to generate cardiovascular patches [83,90].

Recently, QDs, which are luminescent semiconductor nanocrystals with sizes ranging from 1.5 to 10.0 nm, have been employed in numerous biological and medical applications. QDs being photostable has become an ideal candidate for multi-colored imaging and scrutinizing various processes in living cells. Protein labeling methods inside the cells are necessary to study in vivo their localization, movement, interactions, and microenvironments. Often, fusion proteins are created with a GFP tag, but use is limited due to the deterioration of signal, expense of molecular constructs, and the requirement for viral delivery in some cell types [74]. The unique optical and spectroscopic properties of semiconductor QDs offer an alternative to fluorescence-based applications. QDs have a broad excitation profile with narrow, symmetric emission peaks so that mixtures of different sized QDs permit simultaneous excitation with a single light source and detection of multiple targets [91]. This technology has been used to label fibroblasts to study the treatment of third-degree burns in Wistar rats [92]. Histological studies indicated a significant increase in angiogenesis, the number of cells, collagen synthesis, thickness of skin layers, and ultimately accelerated wound healing in treated samples compared to controls. The authors concluded that simultaneous fibroblast therapy and QDs accelerated the repair of skin lesions [92], this study could easily be translated into a myocardial healing setting. Nonetheless, it is still to be determined whether QDs change fibroblast function. Another study showed efficient microRNA delivery using functionalized carbon dots for enhanced conversion of fibroblast to cardiomyocytes [93]. The reprogramming of induced cardiomyocytes is of particular significance in regenerative medicine to replace damaged adult/terminally differentiated cardiomyocytes. The authors reported branched polyethylenimine coated nitrogen-enriched carbon dots as highly efficient nanocarriers. These carbon dots were loaded with a microRNAs combo for cardiac reprogramming and provided an efficient and safe delivery system to induce reprogramming of cardiac fibroblasts into cardiomyocytes. This was accomplished without genomic integration and resulted in effective recovery of cardiac function after MI [93].

### 3.3. Liposomes, PGLA, and Polymeric Micelles

Liposomes are sphere-shaped vesicles consisting of one or more phospholipid bilayers. Liposome size, low toxicity, and aptitude to trap both hydrophilic and lipophilic drugs have increased their use as carriers for numerous molecules both by research labs and pharmaceutical industries. In addition, some therapeutics require frequent dosing due to short half-life, low bioavailability, and/or inability to store the drug in the body. Therapeutic loaded liposomes blunt these limitations. At present, several formulations of liposomes are in clinical use, particularly as anti-cancer therapies [94]. Experimentally, liposomal NPs loaded with MI antigens and rapamycin markedly attenuated inflammation after an acute MI, inhibited adverse cardiac remodeling, and improved cardiac function [95]. In a similar study, cardiac troponin-liposomal NPs delivered miR-21 to the ischemic myocardium, this treatment resulted in improved cardiac function and reduced infarct size [96]. The use of long-circulating liposomes has also been demonstrated to improve therapeutic availability and efficacy by protecting cardiac function against MI in vivo [75]. Liposome NPs also have been extensively used as delivery systems to transfect cardiofibroblasts in mechanistic studies [97,98,99].

Often, a combination of NPs is used to superimpose their benefits. For example, L-carnitine-loaded liposomes and PLGA NP formulations were prepared and characterized to test in vitro delivery of carnitine to cardiac fibroblasts [100]. Efficacy was evaluated by metabolomic studies and pathway analysis. Interestingly, while liposome-only formulations released 90% of their content at the end of the first hour, liposome and PLGA combinations maintained a controlled-release profile for 12 h. The dual formulations also improved fibroblast carbohydrate and lipid metabolism [100].

Several reviews have been published on block copolymers and block copolymer micelles [101,102,103]. Block copolymer micelles can increase the solubility of hydrophobic drugs via sequestration into the hydrophobic core. Further, encapsulation provides for a protective coating for the compound from hydrolysis, enzymatic cleavage, or renal clearance [103]. Therefore, researchers have developed different types of micelles with the goal of improving drug delivery and binding capacity. O’Neil and colleagues generated mixed micelles that display enhanced binding to collagen type 1 [104]. Such micelles could potentially be retained within tissues such as the arterial media after administration from drug delivery catheters.

### 3.4. Adeno-Associated Virus

Adeno-associated virus (AAV) is an alternative method to control gene expression in numerous tissues, with predominant cardioselectivity, specifically cardiomyocytes [105]. AAV carrying a small hairpin RNA targeting tissue nonspecific alkaline phosphatase (TNAP) were used to both knockdowns and overexpress TNAP in the left ventricle after MI and in cardiac fibroblasts from naïve mice [106]. Using this delivery system, the authors demonstrated that TNAP knockdown improves cardiac function and diminishes fibrosis, and this effect likely occurs through the cardiac fibroblast. Similarly, fibroblasts transfected with AAV were instrumental to determine that nucleotide-binding oligomerization domain-like receptor 3 (NLRP3) mediation of cardiac fibrosis occurs via cardiomyocytes and not myoFBs [107] and that overexpression of the left-right determination factor (Lefty)1, a novel member of the TGFβ family, significantly attenuates TGFβ1-induced cardiac fibroblast proliferation, differentiation, and collagen production [108]. Overall, these studies highlight AAV as an effective tool to elucidate fibroblast-dependent mechanisms in cardiac fibrosis.

While numerous AAV vectors continue to be studied, and several are successfully used in in vitro fibroblast transfection, the AAV9 serotype has been shown to be more efficient at in vivo myocardial transduction at lower doses than previously studied AAV serotypes. Since a variety of AAV serotypes transduce cardiomyocytes after systemic administration [109,110], the addition of a promoter sequence for the transcription of periostin, a marker of activated fibroblasts and myoFBs, provides a more targeted fibroblast therapy after ischemic cardiac injury. Piras and colleagues generated a 1395-bp periostin mini-promoter to meet the packaging limit of AAV vectors and tested this delivery system in an I/R MI mouse model. For the first time, they demonstrated the use of periostin-specific AAV9 to target cardiac myoFBs [111].

While AAV vectors have been shown to be nonpathogenic and nontoxic in humans [112], AAV9 therapeutic methods for targeting fibroblasts post-ischemic heart injury are limited by the few numbers of in vivo studies. Moreover, the genome capacity of AAV is limited to approximately 5 kb, though this allows most miRNA genes and clusters to be packaged [113]. Certainly, the use of AAV vectors therapeutically post-ischemic heart injury is promising, but much more research is necessary. Particularly, clinical correlation must be considered with the realities of the time between coronary vessel occlusion and reperfusion and how these methods can be incorporated with current standards of care.

## 4. Discussion

The great plasticity of cardiac fibroblasts allows them to respond quickly to the myocardial injury and to contribute to the subsequent cardiac remodeling. Being the most abundant cell type in the heart, and a key participant in the several phases of tissue healing, the cardiac fibroblast is an excellent target for treating cardiac diseases. The development of cardiac fibroblast-specific mouse lines, including lineage tracing and activation status-specific, has led to substantial strides in cardiac fibroblast research. Additionally, the use of novel targeted delivery approaches (Figure 2) like nanoparticles and modified adenoviruses, has allowed researchers to define the developmental origin of cardiac fibroblasts, elucidate their differentiation pathways, and functional mechanisms in cardiac injury and disease. Even though these novel tools led to significant strides in fibroblast research, still they present limitations. For example, further knowledge on fibroblast phenotype and markers during cardiac injury is necessary to develop more efficient and specific fibroblast-specific transgenes; nanoparticles, although highly promising translational delivery tools, are dose-dependent and may invoke toxicity; and adenoviruses are limited by their genome capacity and potential off-targets. Nonetheless, as these new technologies evolve and are tested, even in different settings of disease, they carry great promise for cardiac fibroblast targeted research and therapeutics.

Finally, although an increasing number of signaling pathways have been referenced to play a role in the regulation of cardiac fibroblast functions and differentiation, more studies are required to reveal the details of these regulations, how these signaling pathways crosstalk with each other, and how they modulate the cardiac microenvironment during progression to heart failure.

## Figures and Tables

**Figure 1 cells-11-00402-f001:**
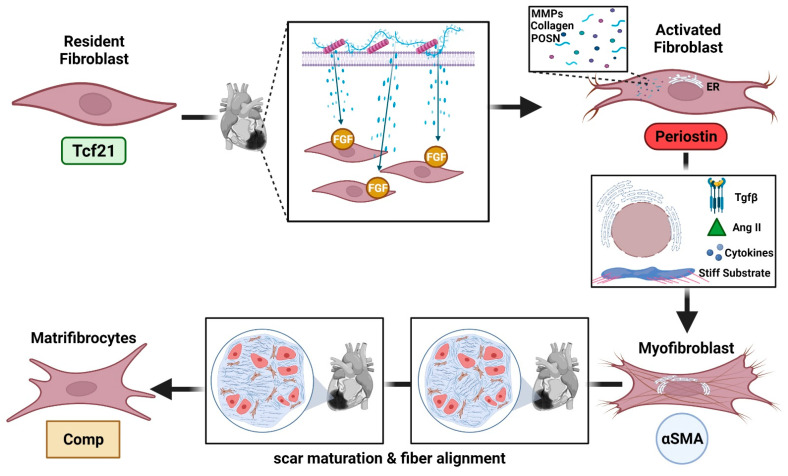
Cardiac fibroblast subpopulations. Four main subpopulations of cardiac fibroblasts have been reported: resident fibroblasts; activated fibroblasts; myofibroblasts; and matrifibrocytes. Upon injury, the resident fibroblasts, identified by the lineage tracing marker Tcf21, differentiate in response to fibrogenic growth factors (FGF) usually stored within the matrix. These activated fibroblasts are phenotypically distinct, present increased adhesion molecules and overexpress several extracellular proteins, such as collagens and periostin (POSN), and matrix metalloproteinases (MMPs). These cells can further differentiate into myofibroblasts, identified by alpha smooth muscle actin (αSMA), upon continuous stress stimuli, such as transforming growth factor beta (TGFβ), angiotensin II (Ang II), cytokines, and increased substrate stiffness (mechanical stimulus). Myofibroblasts display an engorged reticulum endoplasmic (RE) to promote the secretion of extracellular matrix proteins to form the myocardial scar. Long-term remodeling leads to fiber alignment and scar maturation, at this point matrifibrocytes are observed, identified by protein Comp, to stabilize and maintain the mature scar.

**Figure 2 cells-11-00402-f002:**
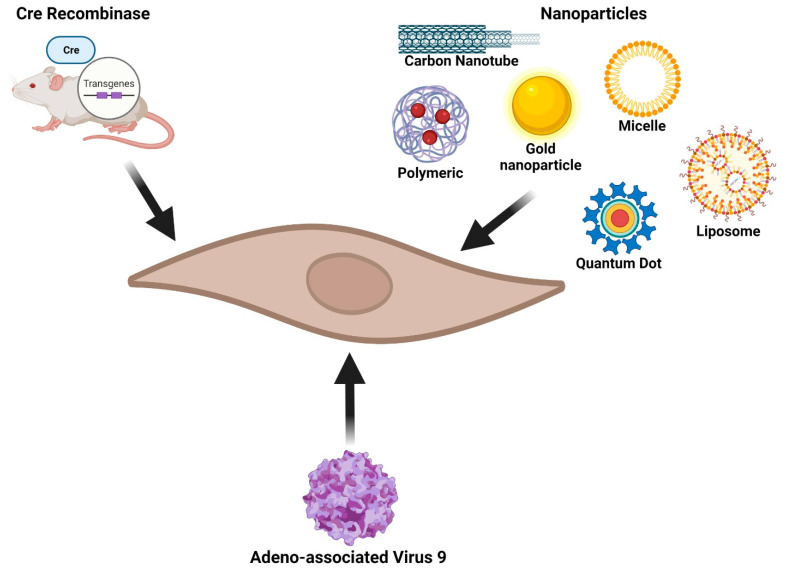
Fibroblast-specific approaches. Cardiac fibroblasts can be specifically targeted via genetic models by combining the use of Cre recombinase and fibroblast-specific promoters (e.g., Tcf21, periostin, collagen 1a1, alpha smooth muscle actin, …). Another popular approach to target fibroblasts is through the use of several types of nanoparticles and liposomes that either encapsulate or are coated with fibroblast-specific antibodies, peptides, or miRNAs. Finally, adeno-associated viruses (AAV) carrying a small hairpin RNA targeting fibroblast-specific proteins have also been shown to be an effective delivery tool in cardiac research.

## Data Availability

Not applicable.

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
