# Peer review of "Novel Techniques Targeting Fibroblasts after Ischemic Heart Injury"

_cells, 2022, doi:10.3390/cells11030402_

Round 1

Reviewer 1 Report

To the authors   - Overall very well written, everything is thoroughly explained.   - In section 'Cardiac fibroblast proinflammatory actions after ischemic injury' the font size is inconsistent, especially in lines 83-91.   - In the section 'Nanoparticles' mesoporous silica nanoparticles (MSNPs) are discussed, but that paragraph is concluded with: 'To our knowledge, MSNPs have not been used in cardiac research'. Since this review focuses on techniques targeting fibroblasts after ischemic heart injury and many other nanoparticles are being discussed, I question the relevance of including the MSNPs in this review.    - A figure giving an overview of either the fibroblast subpopulations (with their function and markers etc.) or fibroblast-specific research tools would add value to this review in my opinion. Right now a lot of information is provided and a figure would allow the reader to recapitulate that information in one glance.

Author Response

Reviewer 1

Below is a response (in italic) point by point to the review:

- In section 'Cardiac fibroblast proinflammatory actions after ischemic injury' the font size is inconsistent, especially in lines 83-91. 

We appreciate your thorough review; we have formatted the text to the correct font size and type.

 - In the section 'Nanoparticles' mesoporous silica nanoparticles (MSNPs) are discussed, but that paragraph is concluded with: 'To our knowledge, MSNPs have not been used in cardiac research'. Since this review focuses on techniques targeting fibroblasts after ischemic heart injury and many other nanoparticles are being discussed, I question the relevance of including the MSNPs in this review.

We described the use of MSNPs in settings of wound healing; however, the reviewer is correct, despite its potential we have found no reports of using MSNPs in cardiac research. We have deleted the section related to MSNPs from this review.  

- A figure giving an overview of either the fibroblast subpopulations (with their function and markers etc.) or fibroblast-specific research tools would add value to this review in my opinion. Right now a lot of information is provided and a figure would allow the reader to recapitulate that information in one glance. Thank you for this suggestion. We have now included a figure describing both fibroblast subpopulations and respective markers. A second figure (as requested by reviewer 3) summarizes fibroblast-specific research tools.

Reviewer 2 Report

I would like to send the following comments about the article you asked to be reviewed   The article represent a comprehensive literature review of the role of fibroblasts in repairing cardiac damage and revises in detail some of the more promising therapeutic targets. Although there are quite a lot of unknown variables in this process, the authors adress some of more important questions, both in acute setting and also in chronic stages.   Also this article adresses the process of cardiac repair after an initial damage in a more fundamental manner, elaborating on the basics and bridging the gap between physiopathology and clinical medicine   There are no further comments and i believe that this review is suitable for publishing.

Author Response

Reviewer 2: Thank you for your comments and review.

Reviewer 3 Report

Comments to Author

The authors summarized the recent reports of cardiac fibrosis after ischemic heart disease. This review is well summarized; however, there are some issues to recommend accepting for publication. These are highlighted in the following points:

The authors may summarize targeting delivery approaches in Table (or figure). It can be easier to understand for readers who are not specialist of drug delivery system.

Author Response

Reviewer 3 response (in italic):

The authors may summarize targeting delivery approaches in Table (or figure). It can be easier to understand for readers who are not specialist of drug delivery system. Thank you for this suggestion.

We have now included a figure summarizing the described fibroblast-specific research tools and delivery systems (new Figure 2).